# New Faecal Calprotectin Assay by IDS: Validation and Comparison to DiaSorin Method

**DOI:** 10.3390/diagnostics12102338

**Published:** 2022-09-27

**Authors:** Vincent Castiglione, Maëlle Berodes, Pierre Lukas, Edouard Louis, Etienne Cavalier, Laurence Lutteri

**Affiliations:** 1Department of Clinical Chemistry, CHU Sart-Tilman, University of Liège, 4000 Liège, Belgium; 2Department of Gastroenterology, CHU Sart-Tilman, University of Liège, 4000 Liège, Belgium

**Keywords:** faecal calprotectin, inflammatory bowel disease, method comparison

## Abstract

Background: The faecal calprotectin (FC) measurement is used for inflammatory bowel disease (IBD) diagnosis and follow-up. The aim of this study was to validate for the first time the new IDS FC extraction device and immunoassay kit, and to compare it with the DiaSorin test in patients with and without IBD. Methods: First, the precision of the IDS assay and its stability were assessed. Then, 379 stool extracts were analysed with the IDS kit on iSYS and compared with a DiaSorin Liaison XL assay. Results: The intra- and inter-assay CVs did not exceed 5%. The stool samples were stable up to 4 weeks at −20 °C. Lot-to-lot comparison showed a good correlation (Lot1 = 1.06 × Lot2 + 0.60; *p* > 0.05). The Passing and Bablok regression showed no significant deviation from linearity between the two methods (IDS = 1.06 × DiaSorin − 0.6; *p* > 0.05; concordance correlation coefficient = 0.93). According to the recommended cut-offs, the IDS assay identified more IBD and irritable bowel syndrome patients than DiaSorin, which had more borderline results (16 vs. 20%, respectively). Conclusions: The IDS faecal calprotectin had good analytical validation parameters. Compared to the DiaSorin method, it showed comparable results, but slightly outperformed it in the identification of more IBD patients and active disease.

## 1. Introduction

Inflammatory bowel disease (IBD) is a chronic disorder that includes Crohn’s disease and ulcerative colitis, which have an increasing prevalence in numerous countries and are a serious health concern. The symptoms include abdominal pain and chronic diarrhea, which are not specific to IBD [1,2]. Therefore, it might be difficult to discriminate IBD from irritable bowel syndrome (IBS), as well as active from inactive IBD, without invasive tests such as colonoscopy. Consequently, there is a need for an efficient, easy to perform, and non-expensive tool for the diagnosis of IBD. It is in that perspective that the measurement of faecal calprotectin (FC) was proposed as a useful biomarker of IBD, as well as for its follow-up [3].

Calprotectin is a 36 kDa heterodimer of two proteins named S100A8 and S100A9, which belong to the calcium-binding proteins family. It is essentially contained in the cytoplasm of neutrophils and is one of its main proteins [4]. Calprotectin plays a role in the innate immune response thanks to its antimicrobial properties, and it is released in the blood or faeces during inflammation. Thus, the calprotectin contained in the stool is essentially coming from neutrophils and is representative of degranulation during intestinal inflammation.

Because active IBD is associated with inflammation, higher levels of FC have been reported in IBD patients, while healthy subjects and IBS patients tend to have normal concentrations [5,6]. FC has consequently been proposed as a biomarker for discriminating IBD from IBS with a good negative predictive value [7,8]. In addition, FC concentration is used for the follow up of IBD because calprotectin release is proportional to the degree of inflammation and hence to the severity of the disease. Recurrent FC measurements allow the prediction of flare-up or the remission of patients with diagnosed Crohn’s disease or ulcerative colitis as its secretion precedes the symptoms [9]. This is particularly useful for the early adaptation of medication doses to prevent the worsening of the disease. Finally, it is a non-invasive test compared to endoscopy with biopsy, which is still the reference method [10]. Indeed, FC is closely correlated with the endoscopic and histological activity of disease [11].

Although several laboratory methods for FC assessment have been developed, including automated immunoassays, ELISA, and point-of-care testing, there is a lack of comparison studies, and to date, none of the evaluated methods has been clearly proven to be better than the others [12]. Moreover, the extraction of calprotectin from the stool samples is a very sensitive pre-analytical step, albeit still necessary, that leads to variability [13]. The current reference method for stool extraction is the weighting method, but it is highly cumbersome and time-consuming. That is why some manufacturers have also developed and commercialized extraction tools to shorten this step.

In this study, we sought to evaluate the new faecal calprotectin assay by IDS (Boldon, UK) on their automated analyser iSYS. In order to do so, we first evaluated the analytical parameters of the IDS kit and its calprotectin extraction device. Then, we compared it to the weighing method and the previously validated DiaSorin (Sallugia, Italy) kit as a reference method and compared their diagnostic performances.

## 2. Materials and Methods

Stool samples were prospectively collected at the university hospital of Liège (Belgium) between January 2020 and May 2021. The faeces were selected regardless of their consistency so that liquid and hard stools were also included. The samples were provided by in- and outpatients of the hospital, consulting at the gastroenterology department. Medical records were checked to identify patients with confirmed IBD (Crohn’s disease and ulcerative colitis). The faeces were collected from the sample’s remnants.

All FC tested on the IDS kit (for validation and comparison as well) were extracted using the IDS EasyCal^®^ collection kit (IS-6040) for pre-analytical processing. The extraction kit consists of a single-use tube prefilled with 2.8 mL of extraction buffer, and a stick with seven grooves that allows the precise collection of 56 mg of stool. After dipping the stick in the faeces sample, it is put back in the tube, vortexed, and centrifuged for 10 min at 3000× *g*. Very hard or liquid stools were treated specifically according to the manufacturer’s recommendations: the highly liquid samples were extracted by directly pipetting 56 µL of stool and placing it in the extraction device buffer, while 100 µL of saline solution was added to the hard faeces for 60 min at room temperature before being dipped as the normal stools were. After extraction, the extracts were stored at −80 °C until measurement.

The IDS calprotectin assay is based on a chemiluminescence sandwich method, with magnetic beads linked to a monoclonal antibody against calprotectin heterocomplex and measured by a second antibody labelled with acridinium. Each test uses 15µL of extracted sample.

### 2.1. Comparison to DiaSorin Assay

For the comparison study, 379 faeces were provided by 194 (51.2%) men and 185 (48.8%) women, whose mean age was 45 (±18) years old. These samples were also used to perform a lot-to-lot comparison of the IDS assay (lot numbers: IS-6000E 5306 and 5372).

The method comparison to the DiaSorin kit was performed by extracting and measuring the 379 samples by two different methods in parallel: the samples were extracted according to the gold standard, e.g., the weighting method, and were tested with the DiaSorin kit on Liaison XL. At the same time, a second aliquot was taken using the IDS EasyCal^®^ collection kit and then tested on the IDS iSYS, as mentioned above.

The DiaSorin assay was selected because, like the IDS one, it also is a CLIA method, using beads with a monoclonal antibody labelled with isoluminol. Both manufacturers recommend using the following cut-off for patient classification: ≤50 µg/g: IBD negative, 50–120 µg/g: borderline (meaning the patient should be re-evaluated again), and ≥120 µg/g: IBD positive. In addition, another positivity cut-off providing better specificity at 250 µg/g was tested [14,15]. Furthermore, the DiaSorin method has previously been compared in several published articles; hence, we considered it as a reference method [13,16,17,18,19].

### 2.2. IDS Method Validation

Four quality controls and 5 human faecal samples were used to establish the inter- and intra-assay variability. The samples were extracted in pentaplicate and stored at −20 °C, and then, each of the 5 aliquots was measured for five days. Concentrations ranging from low (39.6 µg/g) to high (1545.8 µg/g) were selected in order to cover the clinically relevant concentrations.

The stability at 2–8 °C and −20 °C was evaluated using three aliquots of 4 patients’ stool after homogeinization, then extracted and measured in triplicate each time. The short term 2–8 °C storage was tested for up to 7 days with stools of various levels included in the clinical ranges (6.7, 74.2, 164.4, and 1020.2 µg/g), while the frozen (−20 °C) aliquots were tested for up to 14 weeks at the following concentrations: 42.0, 79.2, 133.7, and 707.6 µg/g.

### 2.3. Statistical Analyses

The statistical analyses were performed with MedCalc (version 12.7.70 MedCalc Software Ltd. (Osten, Belgium)). The results of the IDS and DiaSorin methods were compared with the Passing and Bablok regression and the concordance correlation coefficient for the quantitative results and with the Cohen’s kappa for the results classification. The stability was tested with the Wilcoxon paired test.

## 3. Results

### 3.1. Method Validation

The intra-assay variability was assessed on the samples extracted and tested in pentaplicate for 5 days and ranged from 1.8% to 3.0%. The inter-assay variability varied from 2.5% to 4.1% for the tested concentrations (39.6 to 1545.8 µg/g) (Table 1).

The stool samples were stored in the fridge for up to 7 days and in the freezer for up to 4 weeks before analysis. In the fridge (2–8 °C), the aliquots were stable for only 2 days and could not be kept at 2–8 °C for up to 4 days (recovery ≤ 90%). However, the faeces were stable at −20 °C for up to 4 weeks (Figure 1). The Wilcoxon test for paired samples was used for the comparison with the baseline.

The lot-to-lot comparison provided comparable results and no significant deviation from linearity (Lot1 = 1.06Lot2 + 0.60; *p* > 0.05) (data not shown).

The mean time for the FC extraction using the weighting (gold standard) method was 4′30″, while the extraction device required 1′30″ (the time measured during the extraction of 29 samples). Overall, the technicians performing the extraction were satisfied with the use of the extraction device, considering it easier to use compared with the weighting method (Appendix A).

### 3.2. Diagnostic Performance

Stool samples were collected from a total of 379 patients. Among them, 226 (59.6%) had a confirmed IBD diagnosis, 120 (31.7%) had no IBD (IBS or not), and 33 (8.7%) had no defined diagnosis yet. Amongst the IBD patients, 156 (69%) had Crohn’s disease, and 70 (31%) had ulcerative colitis. Many IBD patients had normal FC results because they were already under medical therapy (*n* = 104, 46.0%).

The Passing and Bablok regression for the method comparison between the IDS kit and the DiaSorin kit showed a good correlation (R^2^ Spearman: 0.933; concordance correlation coefficient: 0.932) and no significant difference (IDS = 1.06DiaSorin − 0.6; *p* > 0.05) (Figure 2).

The patients were classified into three categories, from negative to positive, including a borderline range, using the manufacturer’s cut-off. The classification’s agreement between both methods was very good according to the Cohen’s kappa coefficient (K = 0.87); yet, the IDS method allowed the classification of fewer borderline data points (*n* = 59, 15.6%) compared to that of DiaSorin (*n* = 73, 19.3%) (Table 2). Hence, using the IDS method, more patients were immediately classified as positive or negative rather than borderline. Only one IBD patient provided a positive FC result with the DiaSorin method but was negative with the IDS method. It was the only highly discordant result in our study. When testing the 250 µg/g cut-off, widely used among physicians to differentiate active from inactive IBD [14,15], quite similar results were observed: despite a very good agreement (K = 0.80), the IDS assay had fewer borderline data points than that of DiaSorin (Table 3).

## 4. Discussion

In this study, the new IDS assay for faecal calprotectin was compared to the DiaSorin kit, which was considered as the reference method, and was analytically validated; its diagnostic performance was evaluated for the first time to our knowledge.

### 4.1. Method Validation

The IDS FC assay provided results with good precision and low intra- and inter-run variability, all being < 5%, regardless of the tested concentration (from 39.6 to 1545.8 µg/g), and the nature of the sample (native or quality control). As a high intra-individual variability was described by Cremer et al. [20], no consensus has been published to date by any scientific committee about the limits in variability to achieve [21,22]. However, these results can be largely accepted, considering that the coefficient of variation was far below 15%, especially on the stool extract [23,24,25,26].

A particularly important parameter to keep in mind regarding repeatability in faeces analysis, is the pre-analytical extraction step [12,13,19]. In our study, the variability and stability were assessed on different stool extracts, meaning that the variability accounted for both the assays and the extraction. The samples were extracted five times for the precision assessment and three times for the stability assessment using the EasyCal^®^ device. The low coefficients of variation show that the variation due to this preparation step was also very limited, whatever the FC level [13]. Actually, Cremer et al. showed that the variability of FC measured in different extracts from the same stools was low [20]. Although some authors reported that the commercial extraction device led to higher or lower results [16,27], it is probably more repeatable than the weighing method thanks to the simplified process and because it reduces the error risk from human manipulation, in addition to being less time-consuming. Indeed, the extraction protocol using EasyCal^®^ (or other similar devices) requires no cumbersome weighting of the pristine stool sample, the calculation of the buffer to add, and the further homogenisation. Altogether, in our experience, the collection kit proved to be three times faster than the usual weighing method.

Even if intact stool samples may be more stable than protein extracts [28], extracted proteins are less cumbersome to store than faecal samples. FC extracts stored in the fridge (2–8 °C) were only stable for up to 2 days, while they could be stored at −20 °C for up to 4 weeks without significant variation. These results are indeed in accordance with other studies regarding the good conservation at −20 °C [13,29] as well as the lack of stability in the fridge [30,31].

### 4.2. Diagnostic Performance

As has been previously discussed, there can be a wide variation in the results provided by different FC methods, and there is no consensus to date for selecting one manufacturer’s kit over another, even though the ELISA and CLIA tests should be preferred over point-of-care testing because of their better analytical performances, allowing for a better follow-up [16,27,32,33,34]. Nevertheless, the DiaSorin kit has been evaluated by several comparison studies. Overall, these studies reported a good agreement of the DiaSorin assay compared to the others; that is why we selected this method as a reference in the absence of an actual proper gold standard [13,16,17,18,19]. Oyaert et al. found that the DiaSorin and Inova methods had the best agreement amongst six different FC tests and that they had the lowest CVs [19]. Hence, other authors also selected the DiaSorin test as the reference method in a comparison study [17]. In addition, the cut-offs recommended by the manufacturers are the same for IDS and DiaSorin, allowing a better comparison between the patients’ classifications. The correlation between quantitative and qualitative (patients’ classification) results between both methods was good, despite the difference of the reagents used. The results of the concordance correlation coefficient were moderate but considering that this statistical test is more demanding than the Spearman correlation and that we are dealing with stool samples that need to be extracted, these results can be regarded as satisfactory. This good correlation and similar cut-off of the IDS and DiaSorin methods could partly be due to the fact that they both use the CLIA method with antibodies against recombinant calprotectin.

A bias of this study could that concern the diagnostic performance is that we used two different extraction methods for the comparison. However, we used the weighting method for the DiaSorin FC measurement, which still holds the title of reference method. Hence, we compared the full IDS extraction and measurement kit to the DiaSorin method with a gold standard extraction. Yet, the IDS method still provided slightly better results regarding patients’ classification.

Even though the classification comparison was very good (Cohen’s kappa), there were some differences in considering each patient separately. The IDS method obtained slightly fewer results classified as borderline compared with the DiaSorin method, with both using 120 or 250 µg/g cut-offs, resulting in more positive or negative results. Amongst the patients with borderline results with the DiaSorin method, but positive with IDS (*n* = 10), 9 out of 10 had a confirmed IBD diagnosis according to the clinical data. Conversely, using the manufacturer’s cut-offs, only four of the six borderline results with IDS, but positive with DiaSorin, were clinically confirmed to be IBD patients. Quite similarly, using the 250 µg/g positivity criteria, only five of the eight patients providing borderline results with IDS, but positive with DiaSorin, indeed had IBD. Hence, more IBD patients were classified as positive with IDS rather than with DiaSorin, which is probably more accurate clinically, as shown by the concordance with the clinical data. The IDS method would then allow the reduction of the number of repeated stool collections thanks to the lower rate of borderline results [12]. This also strengthens the recommendation that a patient should receive a follow-up with the same method. However, these results should be validated in a longitudinal study.

Whereas seven patients with acute infectious colitis, a well-known cause of very high FC levels, were identified and excluded from the classification comparison, a limitation of the patient’s classification in our study is that we had no information about the IBD patient’s disease activity or remission. Owing to this lack of information, some patients with adequately treated IBD may have low FC measurements. Inversely, non-IBD patients may have high FC due to other pathologies, such as intestinal and rheumatologic disorders [35,36,37]. For instance, gastro-intestinal bleeding, colorectal carcinoma, or adenoma have therefore been associated with an FC increase [38,39]. Overall, the two tests were statistically comparable, with very few differences.

Similarly, there is a lack of an internationally standardized calibrator, which is definitely required in order to improve reproducibility between different methods, as well as between laboratories. Because of this, the FC levels of a patient should always be followed in the same lab, and the results from different labs cannot be used for predicting relapse or remission. This discrepancy in results is certainly due to the fact that the antibodies used by the different methods (monoclonal or polyclonal, mouse or avian antibody, human or recombinant calprotectin), probably bind different parts of the calprotectin heterodimer, leading to these differences. Hence, it is still recommended that each patient has a follow-up with the same method [12]. Several mass spectroscopy methods have been developed to measure human calprotectin in several biological fluids, including saliva or bronchoalveolar lavage fluid. It allowed the discrimination of different forms of calprotectin, such as the 1008A and 1009A intact subunits, or post-translationally modified forms after oxidation, degradation, or polymerisation [40,41,42]. Of note, Edwards et al. showed that some ELISA testing underestimated the calprotectin levels after its oxidation and digestion by the neutrophil’s protease [43]. Hence, mass spectroscopy could help in standardizing the immunochemical kits and gaining a better understanding of the variety of calprotectin present in faeces.

## 5. Conclusions

To our knowledge, this is the first study reporting faecal calprotectin validation and comparison with the IDS kit. The IDS FC had good analytical validation parameters. Compared to the DiaSorin method, the IDS calprotectin assay showed comparable results but slightly outperformed it in the identification of more IBD and IBS patients, rather than borderline results. The use of the extraction device proposed by the manufacturers is user-friendly, less time-consuming and still repeatable. As it is the first study on the IDS assay, further studies are required in order to validate these results. Further harmonization of the FC assay will allow more interchangeability between labs and the establishment of better cut-offs, such as for the discrimination between disease activity and remission.

## Figures and Tables

**Figure 1 diagnostics-12-02338-f001:**
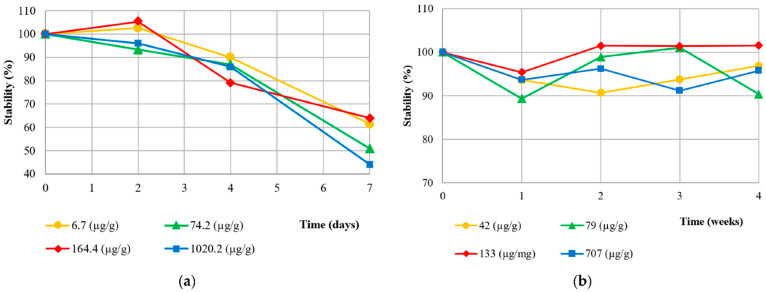
Stability of native stool samples at 2–8 °C (**a**) and −20 °C (**b**).

**Figure 2 diagnostics-12-02338-f002:**
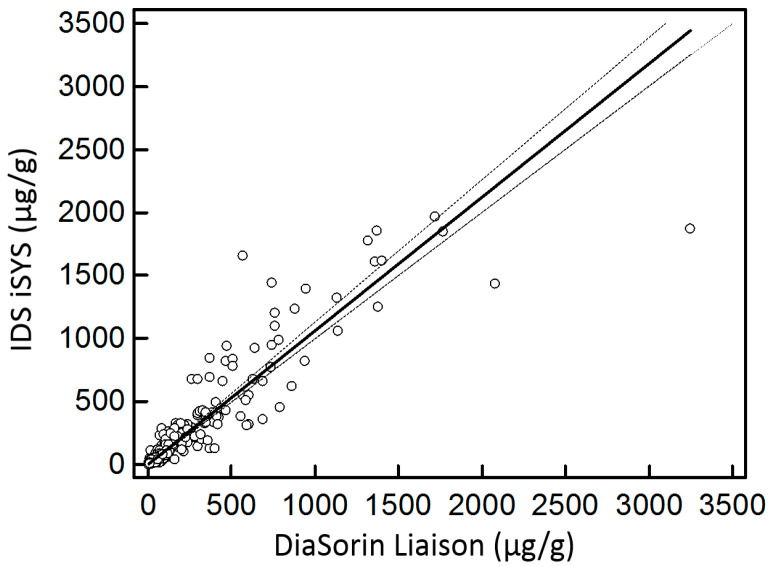
Passing and Bablok regression for FC measurement.

**Table 1 diagnostics-12-02338-t001:** Repeatability of the IDS method.

Concentration (µg/g)	Nature	Intra-Assay CV (%)	Inter-Assay CV (%)
39.6	Sample	1.9	2.6
69.1	Sample	3.0	3.5
72.8	Quality control	2.9	3.6
76.7	Quality control	3.0	3.8
154.1	Sample	1.8	2.5
692.7	Sample	2.0	3.4
776.1	Quality control	2.1	3.5
780.9	Quality control	2.9	4.1
1545.8	Sample	1.9	3.1

**Table 2 diagnostics-12-02338-t002:** Classification of patients according to the recommended 50–120 µg/g cut-offs.

			IDS iSYS	
		Negative (*n* = 203, 53.6%)	Borderline (*n* = 59, 15.6%)	Positive (*n* = 117, 30.9%)
DiaSorin Liaison XL	Negative (*n* = 192, 50.7%)	183	9	0
	Borderline (*n* = 73, 19.3%)	19	44	10
	Positive (*n* = 114, 30.1%)	1	6	107

*n* = 379. Negative < 50 µg/g; borderline 50–120 µg/g; positive > 120 µg/g. Cohen’s kappa classification agreement: 0.87, *p* < 0.001.

**Table 3 diagnostics-12-02338-t003:** Classification of patients according to the recommended 50–250 µg/g cut-offs.

			IDS iSYS	
		Negative (*n* = 203, 53.6%)	Borderline (*n* = 98, 25.9%)	Positive (*n* = 78, 20.6%)
DiaSorin Liaison XL	Negative (*n* = 190, 50.4%)	183	8	0
	Borderline (*n* = 112, 29.6%)	20	82	10
	Positive (*n* = 76, 20.1%)	0	8	68

*n* = 379. Negative < 50 µg/g; borderline 50–250 µg/g; positive > 250 µg/g. Cohen’s kappa classification agreement: 0.80, *p* < 0.001.

## Data Availability

Not applicable.

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
