# Peer review of "New Faecal Calprotectin Assay by IDS: Validation and Comparison to DiaSorin Method"

_diagnostics, 2022, doi:10.3390/diagnostics12102338_

Round 1

Reviewer 1 Report

The manuscript entitled “New faecal calprotectin assay by IDS: validation and compari- 2 son to DiaSorin method’ is quite interesting to diagnosis of e inflammatory bowel disease (IBD). Authors have developed new IDS FC extraction device and immunoassay kit and compared Comparison to DiaSorin assay. Authors have to mentioned that proper methodology for new IDS Calprotectin assay and what is novelty and significance.

Author Response

We greatly thank the reviewer for their comments and valuable remarks. The methodology section has been clarified according to the recommendation. Other improvements made in the text according to both reviewers’ comments are marked with a red line. We hope that the reviewer finds these modifications suitable.

Reviewer 2 Report

Authors compare the new IDS assay to the standard assay DiaSorin in the measurement of faecal calprotectin, which is used in IBD diagnosis and follow-up. Results show similar results in performance (with slightly better results in the diagnosis of IBD patients in active disease) and good repeatability and robustness of the new IDS assay.

A few questions here below:

In the methods section, it reads as if the same samples were gone through the same method, one after another, or it was an aliquote of the same sample? Please clarify.

Has the IDS method previously been compared against other methods? If additional potential new methods had been identified and also considered reliable, such methods should also have been compared to IDS, especially the ones shown to outperform or perform similarly to DiaSorin.

Why did the authors not use the diagnostic performance between IBD versus non-IBD patients and used AUC to compare the results? Identifying the cutoffs from the ROC analysis for the results from IDS could be helpful to compare (the one that gives the largest sensitivity + specificity). 

Perhaps those thresholds are fined-tuned to DiaSorin, and should not be widely used for other methods, which tend to give smaller values of FC than other methods. 

Please find below some suggestions and typos spotted: 

Abstract: did not exceed, not exceeded

is used for inflammatory, not for the inflammatory

Lot-to-lot

Correlation value not presented? 

Line 52: Point-of-care

Line 53: to-date

Line 54: from stool samples

Line57: have developed and commercialized

Line 63: Performances

Line 77: highly liquid samples

Line 79: minutesat, minutes at

Line 88: lot-to-lot

Line 93: weighting method

Line 96: manufacturers recommend 

Line 96: for patient classification

Line 105: each of the 5 aliquots, or the 5 aliquots

Line 106: space before micro gram/g extra, please keep consistency

Line 114: MedCalc

Line 133 : Lot-to-lot

Line 155: fewer borderline data points

Line 160: active from inactive

Line 161: fewer borderline data points

Line 163, Table 2 Title: patients

Line 167, Table 3 Title: patients  

Table 3: Negative (n=190

Line 186: stability assessment

Line 215: patients’ classification

Line 216, 225: patients’ classification

Author Response

The authors greatly appreciate the reviewer’s comments that helped us to improve the manuscript. Every remark has been taken in account and modified in the manuscript. In addition, here are the answers to major comments. We hope that the reviewer finds these modifications satisfactory.

Concerning the methodology, two aliquots were taken from the same stool sample at the same time. The following section was clarified as follow: “The method comparison to the DiaSorin kit was performed by extracting and measuring the 379 samples by two different methods in parallel: the samples were extracted according to the gold standard, e.g. weighting method, and were tested with the DiaSorin kit on Liaison XL. At the same time, a second aliquote was taken using IDS EasyCal® collection kit, and then tested on the IDS iSYS as mentioned above.”

To our knowledge, there is no publication on the IDS method to-date. This was checked again this 20th September 2022 on Pubmed with the following words: IDS fecal calprotectin. Only one result was found, but was out of scope. That is why we stated “this is the first study reporting faecal calprotectin validation and comparision with the IDS kit.”

Concerning cut-off, we unfortunately had no data about IBD activity. Hence, predicting new cut-off thanks to a ROC analysis might be biased. In addition, we had no population to test and verify these new cut-off. In opposition, we used the cut-off recommended by both manufacturers.

Regarding the 250µg/g cut-off, it has indeed been mainly described with the Buhlmann method, which is known to be different from DiaSorin. However, in a recent study that we are about to submit for publication, we tested the ROC analysis of DiaSorin method to discriminate inactive from active IBD (data below). A cut-off at 248µg/g showed specificity >90%, in agreement with the published recommendations. Please see the data in the attached file.

Finally, we greatly thank the reviewer for the minor corrections, including language and typos. A special emphasis has been given to the correlation value that had been previously been calculated with the Spearman correlation. In addition, we used the concordance coefficient correlation that is a more demanding test (ccc = 0.93, moderate correlation). Hence, a few sentences have been added to the methodology, results and discussion to briefly discuss it.
